# Synergic Effect of the Antimicrobial Peptide ToAP2 and Fluconazole on *Candida albicans* Biofilms

**DOI:** 10.3390/ijms25147769

**Published:** 2024-07-16

**Authors:** Jhones do Nascimento Dias, Fabián Andrés Hurtado Erazo, Lucinda J. Bessa, Peter Eaton, José Roberto de Souza de Almeida Leite, Hugo Costa Paes, André Moraes Nicola, Ildinete Silva-Pereira, Patrícia Albuquerque

**Affiliations:** 1Laboratory of Molecular Biology of Fungi, University of Brasilia, Brasilia 70910-900, Brazil; jhonesnd@gmail.com (J.d.N.D.); fahejml@gmail.com (F.A.H.E.); 2LAQV/REQUIMTE, Department of Chemistry and Biochemistry, Faculty of Sciences, University of Porto, 4169-007 Porto, Portugal; lbessa@egasmoniz.edu.pt (L.J.B.); peaton@lincoln.ac.uk (P.E.); 3The Bridge, School of Chemistry, University of Lincoln, Lincoln LN6 7TS, UK; 4Center for Research in Applied Morphology and Immunology, NuPMIA, University of Brasilia, Brasilia 70910-900, Brazil; jrleite@unb.br; 5Faculty of Medicine, University of Brasilia, Brasilia 70910-900, Brazil; hugopaes@unb.br (H.C.P.); amnicola@unb.br (A.M.N.)

**Keywords:** *Candida albicans*, antifungal drugs, antimicrobial peptides, synergism, fluconazole, biofilms

## Abstract

*Candida albicans* is one of the agents of invasive candidiasis, a life-threatening disease strongly associated with hospitalization, particularly among patients in intensive care units with central venous catheters. This study aimed to evaluate the synergistic activity of the antifungal peptide ToAP2 combined with fluconazole against *C. albicans* biofilms grown on various materials. We tested combinations of different concentrations of the peptide ToAP2 with fluconazole on *C. albicans* biofilms. These biofilms were generated on 96-well plates, intravenous catheters, and infusion tubes in RPMI medium at two maturation stages. Scanning electron microscopy and atomic force microscopy were employed to assess the biofilm structure. We also evaluated the expression of genes previously proven to be involved in *C. albicans* biofilm formation in planktonic and biofilm cells after treatment with the peptide ToAP2 using qPCR. ToAP2 demonstrated a synergistic effect with fluconazole at concentrations up to 25 µM during both the early and mature stages of biofilm formation in 96-well plates and on medical devices. Combinations of 50, 25, and 12.5 µM of ToAP2 with 52 µM of fluconazole significantly reduced the biofilm viability compared to individual treatments and untreated controls. These results were supported by substantial structural changes in the biofilms observed through both scanning and atomic force microscopy. The gene expression analysis of *C. albicans* cells treated with 25 µM of ToAP2 revealed a decrease in the expression of genes associated with membrane synthesis, along with an increase in the expression of genes involved in efflux pumps, adhesins, and filamentation. Our results highlight the efficacy of the combined ToAP2 and fluconazole treatment against *C. albicans* biofilms. This combination not only shows therapeutic potential but also suggests its utility in developing preventive biofilm tools for intravenous catheters.

## 1. Introduction

Candidiasis is one of the most common opportunistic fungal infections, especially in the healthcare environment. This disease can be caused by different species of *Candida*; however, infections with *Candida albicans* are the most common [1]. *C. albicans* is a commensal organism that can be found in different parts of the human body such as the gastrointestinal and reproductive tracts, mouth, and skin without causing damage to the host [2]. However, in response to some stimuli, such as certain types of immunodeficiency, a disturbance in the microbiota balance, or the presence of lesions in the epithelium, this fungus can become an important human pathogen [3].

The most common forms of candidiasis are vulvovaginal, oral, and esophageal, and in more severe cases, this pathogen can reach the bloodstream and spread to other organs [4]. According to the World Health Organization (WHO), patients with invasive candidiasis can be hospitalized for weeks, sometimes even months, with a mortality rate ranging from 20% to 50%. In light of these reports, the WHO has designated this species as a priority fungal pathogen for public health development and action [5].

*C. albicans* has several well-characterized virulence factors which can contribute to pathogenesis [6]. These include the ability of *C. albicans* to transition between different morphological forms [7], its production of adhesins [8], its metabolic plasticity, its secretion of enzymes, and its ability to form biofilms [9]. Among these, the last one is a major concern in clinical practice [10].

Biofilms are microbial communities strongly adhered to a surface and surrounded by an extracellular matrix [11]. In the clinical setting, these communities pose an important threat to human health due to their intrinsic high resistance to antimicrobials and to the immune system [12,13]. Biofilms are closely associated with the long-term use of many medical devices such as catheters, pacemakers, and implants [14], and *C. albicans* is considered one of the most common microorganisms causing device-associated infections [15].

Azoles are the most used antifungal class in the treatment of candidiasis. Among the advantages of this group of antifungals is the possibility of oral administration, the low costs of treatment, their limited toxicity, and their broad spectrum of action [16,17]. However, *C. albicans* biofilms present an intrinsic resistance to azoles, demanding the use of other more toxic and/or expensive antifungals to treat biofilm-associated infections and reinforcing the need to develop better antifungal therapies [10,18].

In the last two decades, antimicrobial peptides (AMPs) have been attracting increasing attention in the development of new therapeutic strategies to treat infectious diseases [19,20]. AMPs are small molecules present in several organisms and are an effective part of their innate immune response against pathogens [21]. In general, they are cationic, amphipathic, and have a broad spectrum of activity against various microorganisms [22,23,24]. Their main mechanism of action is the physical damage of microbial membranes in a receptor-independent way, which is less prone to the development of antimicrobial resistance [25]. However, among the obstacles in the clinical development of therapies using AMPs is the toxicity of some AMPs to mammalian cells [26]. A strategy to overcome this problem is the use of AMPs in combination with other antifungals, reducing their working concentrations to safe levels and avoiding the problem of resistance development [27].

In this context, we evaluated the combined in vitro and in vivo action of the scorpion venom-derived antimicrobial peptide ToAP2 and NDBP-5.7 [28,29] with Fluconazole or Amphotericin B against *C. albicans* biofilms.

## 2. Results

### 2.1. Synergism between ToAP2 with Antifungals in Different Stages of Biofilm Formation

The impact of combining ToAP2 with either Amphotericin B or fluconazole was assessed during both the early and mature phases of biofilm formation using the SynergyFinder Plus package version 3.10.3 [30]. Our results revealed no significant synergistic effect in the combinations of the peptide with Amphotericin B during both phases of biofilm formation (Figure 1A,B). The mean synergy score was 3.16 (*p*-value = 4.95 × 10^−1^) for the initial phase and 10.52 (*p*-value = 7.32 × 10^−2^) for mature biofilms (Appendix A).

In contrast, the combinations of ToAP2 with fluconazole exhibited a synergistic effect in both phases of biofilm formation. Specifically, combining ToAP2 at concentrations of 12.5 μM or higher with any of the tested concentrations of fluconazole (1.62, 3.25, 6.5 μM) resulted in a reduction of over 50% in the biofilm (Figure 1C). The highest synergy score (44.19 ± 2.02; *p*-value = 8.60 × 10^−23^) was observed at concentrations of 12.5 μM of ToAP2 and 0.52 μM of fluconazole (Appendix A). Similarly, in the mature biofilm, the highest score was achieved with the combination of 100 μM ToAP2 and 0.52 μM fluconazole (synergy score: 59.27 ± 9.46; *p*-value = 7.72 × 10^−4^) (Appendix A). The synergy scores reflect the average excess response resulting from drug interactions. According to this, a synergy score less than −10 indicates an antagonistic interaction between the drugs. Scores from −10 to 10 suggest an additive interaction, while scores greater than 10 indicate a synergistic interaction [31]. Additionally, combining 100 μM of ToAP2 with any concentration of fluconazole resulted in a biofilm reduction of more than 50% (Figure 1D).

### 2.2. ToAP2 Increases the Damage to Membranes of C. albicans Biofilm Cells in the Presence of Fluconazole

We evaluated biofilm damage using both scanning electron and atomic force microscopy. In Figure 2, treatment with 100 µM ToAP2 peptide (Figure 2A) or with 1.08 µM Amphotericin B (Figure 2D) resulted in noticeable regions of wrinkling on the biofilm surface which are indicated by white arrows in the image. On the other hand, the untreated control showed smooth, intact cell surfaces (Figure 2G). In contrast, the biofilms treated with 100 µM of another AMP, NDBP-5.7, showed few effects on biofilm structure (Figure 2B), while those treated with 52 µM Fluconazole (Figure 2C) exhibited mostly no significant difference in comparison to the untreated control (Figure 2G). The white arrow in Figure 2C highlights an area of cell collapse. Interestingly, biofilm damage was increased when ToAP2 (100 µM) was combined with Amphotericin B or with fluconazole (Figure 2E and Figure 2F, respectively).

The AFM analysis not only confirmed the combinatory effects observed in the SEM images but also provided a more detailed assessment of the impact of NDBP-5.7 on *C. albicans* biofilms. The AFM figures once again showed significant changes, including cell collapse and roughening of the membranes in all the treated samples, except for those treated with 52 µM Fluconazole, similar to the results seen from SEM. The evaluation of the membrane roughness by AFM revealed a statistically significant increase in the average surface roughness of the biofilms treated with both ToAP2 and NDBP-5.7 in comparison to the control (Figure 3H, *p* < 0.0001). In contrast, the treatments with Fluconazole or Amphotericin B alone did not alter the surface roughness of the biofilms relative to the control.

### 2.3. ToAP2 Combined with Fluconazole Reduces the Viability of C. albicans Biofilm in Different Materials

In addition to the assays performed in the 96-well plates, we extended our evaluation of the antifungal activity of ToAP2 to the biofilms formed in clinical materials such as intravenous infusion and intravenous catheter tubes (PU catheters). The treatment of the *C. albicans* biofilms in infusion tubes with 50, 25, or 12.5 µM of ToAP2 alone or with 52 µM of Fluconazole alone did not impact the viability of those biofilms. However, a significant reduction in biofilm viability was observed when the biofilms were treated with the same concentrations of ToAP2 in the presence of Fluconazole compared to the control (Figure 4A). The treatment of the PU catheters with ToAP2 or ToAP2 with fluconazole yielded effects consistent with those observed in the intravenous infusion model (Figure 4B).

### 2.4. ToAP2 Alters Expression of Genes Related to C. albicans Virulence Factors

We analyzed the effects of ToAP2 on *C. albicans* cells regarding the modulation in the expression of diverse genes associated with fungal pathogenesis (Figure 5). In the analysis of planktonic cells (Figure 5A), there was a significant upregulation of the transcripts of genes related to fungal cell adhesion to the surface of host cells and biofilm formation such as HWP1, which encodes a cell wall protein involved in hyphae formation and the ALS family genes (ALS1, ALS3, and ALS5), which encode adhesion surface glycoproteins [32,33,34,35,36], as well as fungal efflux pump-mediated resistance (CDR1, CDR2, and MDR1) [37,38]. Interestingly, we observed a downregulation of transcript accumulation for ERG11, a key gene in the ergosterol biosynthesis pathway and, in turn, in azole drug resistance [39,40,41,42].

Although the accumulation of transcripts for most of the evaluated genes was not affected by the ToAP2 treatment in the *C. albicans* biofilms (Figure 5B), an upregulating effect was observed in the ALS1 and ALS5 genes at both concentrations of ToAP2 tested (25 µM and 50 µM).

## 3. Discussion

The ability of *C. albicans* to form biofilms on medical devices is a significant source of nosocomial infections associated with elevated morbidity and mortality. This is exacerbated by the high intrinsic resistance of biofilms to antifungals, such as azoles and Amphotericin B, compared to planktonic cells [43,44]. Additionally, many available antifungals are known for their toxicity to mammalian cells, especially at higher concentrations, limiting their clinical utility [45]. To address this issue and also mitigate the development of antimicrobial resistance, combined antifungal therapies have been proposed [14]. While we have previously observed a synergistic effect in planktonic cells [29], the combination of ToAP2 and Amphotericin B demonstrated a distinct impact on biofilms. An assessment of these combinations during the adhesion phase showed minimal differences in comparison to the effects of individual drugs. In the mature biofilms, although certain combinations displayed increased antifungal activity, no statistically significant difference was observed compared to individual compounds, suggesting a neutral effect on the biofilm. These findings suggest a potential similarity in the mechanisms of action of ToAP2 and Amphotericin against biofilms [46].

On the other hand, the outcomes of combining ToAP2 with Fluconazole appear promising, as this exhibited a synergistic inhibitory effect in both phases of biofilm formation. Remarkably, in the presence of 25 and 12.5 µM of the peptide, even minimal concentrations of Fluconazole were able to reduce the viability of the biofilm in the adhesion phase to less than 50%. In previous work, we demonstrated that these concentrations show very low cytotoxicity against human erythrocytes and murine peritoneal macrophages [28]. In addition, there was also a marked reduction in the mature biofilms when 25, 50, and 100 µM of ToAP2 was used with different Fluconazole concentrations. This finding is particularly interesting considering the intrinsic resistance of *C. albicans* biofilms to Fluconazole [47,48]. This suggests a potential avenue for the development of combined therapies with this antifungal, which is especially noteworthy as Fluconazole belongs to one of the main classes of antifungals used in the clinical settings against candidiasis.

SEM and AFM analysis of biofilms treated with ToAP2 and Fluconazole revealed significant structural alterations, suggesting that the peptide also targets the cell membrane—a well-stablished mechanism of action for various AMPs [49]. Similarly, in a study with the peptide P-113, the authors showed that this peptide induced protuberances in the *C. albicans* biofilm filaments [50]. Interestingly, Fluconazole alone exhibited no discernible impact on biofilm morphology. However, when combined with ToAP2, we observed a notable reduction in biofilm size alongside morphological changes (cell shrinkage and increased cell roughness), as is evident in both SEM and AFM images. Notably, ToAP2, when administered alone, induced structural changes like the ones observed with amphotericin treatment. The SEM analysis of the *C. albicans* biofilm treated with another peptide, NDBP-5.7, did not reveal any major structural changes, but the AFM analysis revealed an increase in the biofilm surface roughness. The combination of ToAP2 and Fluconazole also revealed an increase in the biofilm roughness. Alone, neither Amphotericin B nor Fluconazole induced changes in the biofilm roughness in comparison to the untreated control.

These results demonstrate the great potential of the combination of Fluconazole and ToAP2 in eliminating biofilms on surfaces such as intravenous catheters, which are closely associated with systemic candidiasis. Cools et al. [51] found similar results with the combination of caspofungin and the peptide HsLin06_18, which significantly reduced *C. albicans* biofilms in intravenous catheters. Additionally, Raman et al. [52] showed that catheters loaded with different polymers containing a β peptide were able to inhibit the growth of *C. albicans* biofilms.

Our findings also suggest that the combined therapeutic potential of Fluconazole and ToAP2 can be leveraged in developing biotechnological tools for coating surfaces to prevent *C. albicans* biofilms. Interestingly, in the presence of ToAP2, the expression of various genes associated with *C. albicans* virulence was altered, as assessed by qRT-PCR. In planktonic cells, genes related to adhesion (HWP1, ALS1, and ALS3) and efflux pumps (CDR1 and MDR1) showed increased expression at different tested concentrations. This phenomenon might imply a response by *C. albicans* to counteract the adverse effects of the therapeutic agent. Conversely, the expression of the ERG11 gene, associated with the ergosterol synthesis pathway, was reduced, suggesting that the peptide targets the plasma membrane, as demonstrated in previous studies [29]. A similar alteration in the expression of ERG11 and ERG5 along with an increase in the accumulation of CDR1 transcripts were observed in the presence of the AMP MAF-1A after two hours of interaction with *C. albicans* [42].

In the evaluation of gene expression in the *C. albicans* biofilms, only the ALS1 and ALS5 genes showed a significant increase in expression. This observation aligns with previous works suggesting that the upregulation of specific adhesin genes plays a significant role in the biofilm formation and pathogenesis of *C. albicans* [53]. Contrasting with our results, Maione et al. [34] observed a significant reduction in the expression of ERG11 and ALS5 in *C. albicans* biofilm in the presence of the WMR peptide. This difference may be related to the biofilm’s exposure time to the peptide, with the researchers maintaining the biofilm’s contact with the peptide for 24 h while our analysis was performed 1 h after the treatment.

In conclusion, the ToAP2 peptide not only impacts *C. albicans* biofilms independently but also in combination with clinically used antifungals. Its effects extend beyond direct implications for *C. albicans* cell viability, encompassing significant changes in biofilm morphology at subinhibitory concentrations across various medical devices. The synergistic action with Fluconazole is particularly noteworthy, presenting a promising avenue for novel therapeutic approaches and the coating of medical devices. This is especially relevant since Fluconazole, a widely employed antifungal in candidiasis treatment, lacks efficacy against *C. albicans* biofilms.

## 4. Materials and Methods

### 4.1. Synthesis of Peptides

The AMPs ToAP2 (FFGTLFKLGSKLIPGVMKLFSKKKER, 3 KDa, net charge: +6) and NDBP-5.7 (ILSAIWSGIKSLF-NH2, 1.43 KDa, net charge: +1) were chemically synthesized by Biomatik using an Fmoc/t-butyl on solid support strategy. The peptide sequences were derived from scorpion gland cDNA libraries and purification and characterization took place as described previously [28].

### 4.2. C. albicans Culture Conditions

*C. albicans* SC 5314 was used in all the experiments. Aliquots from frozen stocks in 30% glycerol were plated on Sabouraud dextrose agar for 24 h at 30 °C. After growth, a colony was selected and inoculated in Sabouraud dextrose broth at 30 °C (200 rpm) for 24 h. After growth, the cells were washed three times with sterile phosphate buffer (PBS) (1000× *g* at 25 °C) and counted using a hemocytometer, then the cultures were diluted to appropriate cell densities according to each experiment.

### 4.3. Antimicrobial Activity of ToAP2 in Combination with Other Antifungals

The antifungal activity of ToAP2 against *C. albicans* SC 5314 biofilms was performed as previously described [54]. An inoculum of 1 × 10^5^ *C. albicans* cells in 100 μL of RPMI 1640 medium was added to 96-well polystyrene microplates and incubated at 37 °C for 4 or 24 h (early and mature phases respectively). After that, biofilms were washed three times with PBS to remove non-adherent cells, received fresh medium with different concentrations of ToAP2 combined with Amphotericin B or Fluconazole, and the plates were incubated at 37 °C for 24 h. Cell viability was evaluated using the Alamar Blue (ThermoFisher, Waltham, MA, USA) reagent and fluorescence was read using a SpectraMax^®^ M plate reader (Molecular Devices, LLC, San Jose, CA, USA), with excitation at 550 nm and emission at 585 nm. To capture the combinatorial multidose–response effects, drug synergism was assessed using SynergyFinder Plus R package [31]. Fluorescence intensity readings were normalized to the average of the control wells on the same plate to determine relative cell viability values. All cell viability data were transformed to inhibition values. Synergism analysis was performed applying the Zero Interaction Potency (ZIP) model, which captures drug interaction relationships by comparing the change in the potency of the dose–response curves between individual drugs and their combinations [31,55]. Plots to visualize synergy maps for two-drug combinations were produced in R using the package SynergyFinder Plus 3.10.3. The summary synergy scores represent the average excess response due to drug interactions. In general, a synergy score less than −10 indicates antagonistic interaction between the drugs. Scores from −10 to 10 suggest an additive interaction, while scores greater than 10 indicate a synergistic interaction [31].

### 4.4. Scanning Electron Microscopy (SEM) and Atomic Force Microscopy (AFM)

SEM images were made as previously described [56]. To generate *C. albicans* biofilms, 10^6^ yeast cells were inoculated in RPMI medium on 24-well plates containing sterile coverslips and incubated at 37 °C for 24 h. After biofilm formation, the coverslips were gently washed with PBS and the biofilms were treated for 24 h with ToAP2 (100 µM), NDBP-5.7 (100 µM), Amphotericin B (1.08 µM), Fluconazole (52 µM), or with combinations of ToAP2 and the two antifungals. An untreated group was used as a control. After the treatments, the samples were fixed with 2.5% glutaraldehyde in sodium-cacodylate buffer, washed with ultrapure water, and air-dried at room temperature. Samples for SEM analysis were coated with Au/Pd before analysis and imaged using an FEI Quanta 400 SEM in high vacuum, at 10 KeV accelerating voltage. Samples for AFM were scanned without metal coating using a TT-AFM from AFM Workshop in vibrating mode, using a 50 × 50 × 17 µm scanner using ACT probes with resonant frequency of approximately 300 kHz (Applied NanoStructures, Inc., Mountain View, CA, USA). Micro-roughness was evaluated using the Gwyddion 2.40 software.

### 4.5. Effect of the ToAP2 Peptide on Biofilms in Different Clinical Devices

The action of the ToAP2 peptide on an in vitro catheter model of *C. albicans* biofilms was performed as previously described [57]. Before the inoculum, fragments of intravenous catheters and tubes for intravenous infusion (1 cm) were incubated in 24-well plates with fetal bovine serum for 12 h. After that, the catheters were incubated in the presence of 5 × 10^6^ cells of *C. albicans* for another 24 h at 37 °C in RPMI medium for biofilm formation. The fragments were then transferred to new plates, washed carefully with PBS, and incubated for another 24 h in the presence of ToAP2, Fluconazole, or combinations of the two molecules. Biofilm viability was assessed using the viability reagent Alamar Blue in a SpectraMax^®^ M plate reader, and untreated groups were used as growth control.

### 4.6. Evaluation of C. albicans Gene Expression in Response to AMP Treatment

Control and AMP-treated *C. albicans* cells were subjected to gene expression analysis by qRT-PCR. An inoculum of 2 × 10^7^ planktonic cells of *C. albicans*, treated or untreated with 25 and 50 µM of ToAP2, was incubated at 37 °C with shaking (200 rpm) for 1 h. Following the treatment, fungal cells were washed with PBS and subjected to total RNA extraction.

The conditions used for biofilm formation and AMP treatment were as described before [29]. Briefly, stock cells from *C. albicans* were thawed and cultured in Sabouraud dextrose broth for 18 h at 30 °C with shaking at 200 rpm. Then, cells were washed with PBS and centrifuged (1000× *g* at 25 °C). An inoculum of 1 × 10^6^ cells/mL of *C. albicans* in 100 µL of RPMI 1640 medium was added to 96-well microplates. Plates were then incubated for 24 h at 37 °C to promote full biofilm formation. After the incubation time, biofilms were washed with PBS to remove non-adherent cells. RPMI medium with different concentrations of ToAP2 was added and incubated for 1 h at 37 °C. Then, biofilms were washed and detached from the plates for total RNA extraction to perform sessile cell gene expression evaluation by qRT-PCR.

Total RNA was extracted from *C. albicans* planktonic and biofilm cells using mirVana™ miR Isolation Kit (Invitrogen, Thermo Fisher Scientific, Waltham, MA, USA) according to the manufacturer’s specifications.

RNA samples were quantified using a NanoDrop One^c^ instrument (Thermo Fisher Scientific, Waltham, MA, USA), and their quality was analyzed by electrophoresis on agarose gels. RNA samples were subjected to DNase treatment (Promega, Madison, WI, USA) and were stored at −20 °C. For cDNA synthesis, total RNA was reverse transcribed from 1 µg of total RNA using High-Capacity cDNA Reverse Transcription kit (Applied Biosystems, Waltham, MA, USA) according to the manufacturer’s instructions. The qPCR reactions were performed in triplicate using Fast SYBR Green Master Mix^®^ (Applied Biosystems) and carried out in the 7500 Fast Real-Time PCR System (Applied Biosystem). The genes evaluated were Hwp1, Als1, Als3, Als5, Mdr1, Cdr1, Cdr2, Erg6, Erg11, and the housekeeping control Act1 (Table 1). Changes in transcript abundance in stimulated samples were compared to non-stimulated control group using the 2^−∆∆Ct^ method [58].

### 4.7. Statistical Analysis

For the analysis, three independent experiments were performed in technical triplicate. The multiple group comparisons were conducted with a one-way analysis of variance (ANOVA) followed by Tukey’s post-test. Synergism between the antifungal drugs and AMP combinations was evaluated using the ZIP model from SynergyFinder Plus package 3.10.3 in the R environment. For the qPCR experiments, comparisons were conducted using Student’s *t*-test. A *p*-value < 0.05 was considered to indicate a statistically significant difference. Statistical analyses were carried out with GraphPad Prism software, version 8.

## Figures and Tables

**Figure 1 ijms-25-07769-f001:**
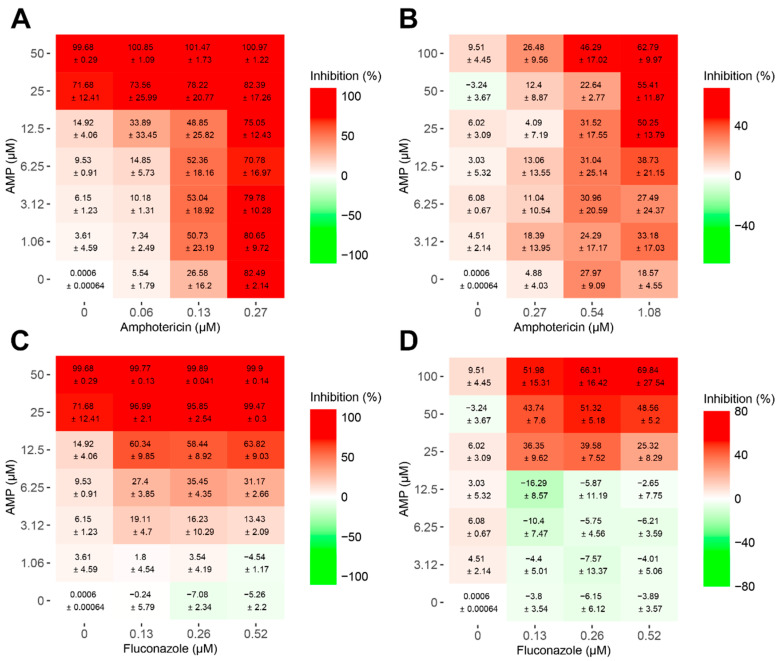
Biofilm inhibition dose–response matrix after treatment with ToAP2 and conventional antifungals. (**A**,**B**) Inhibition of early-phase (4 h) and mature (24 h) *C. albicans* biofilms after 24 h treatment with ToAP2 and/or Amphotericin B, respectively. (**C**,**D**) Inhibition of early-phase (4 h) and mature (24 h) *C. albicans* biofilms after 24 h treatment with ToAP2 and/or fluconazole, respectively. Cell viability was measured by fluorescence with Alamar blue reagent after 2 h of incubation. The viability data were transformed to inhibition using the SynergyFinder Plus R package 3.10.3. The axes represent the concentrations for ToAP2 (rows) and each antifungal (columns). The heatmap scales vary from red, indicating no growth, to green, indicating maximum growth. Data are presented as mean ± standard error of the mean of three independent assays.

**Figure 2 ijms-25-07769-f002:**
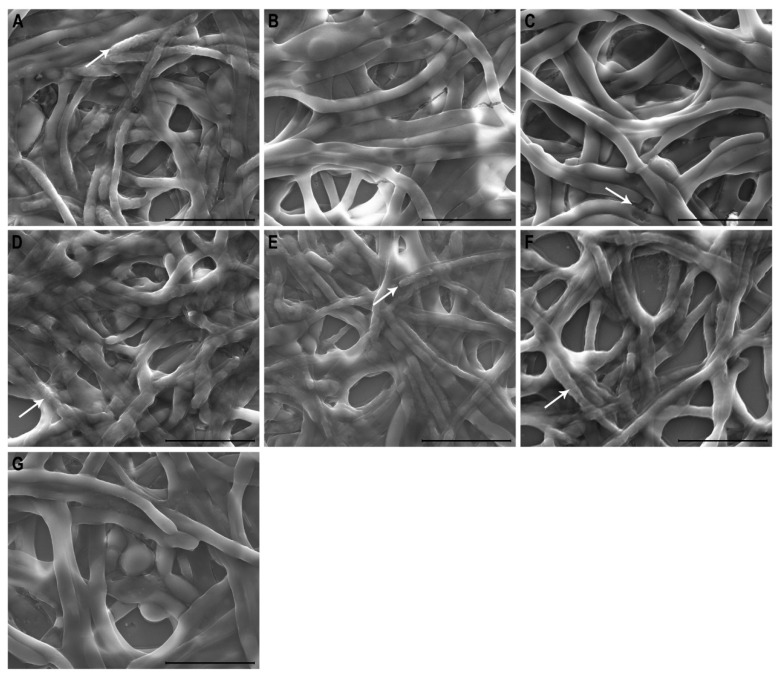
SEM images of *C. albicans* mature biofilms after different treatments. (**A**) Treatment with 100 µM ToAP2, (**B**) 100 µM NDBP-5.7, (**C**) 52 µM Fluconazole, (**D**) 1.08 µM Amphotericin B, (**E**) combination of ToAP2 and Amphotericin, (**F**) combination of ToAP2 and fluconazole, (**G**) untreated control. Scale bar: 10 µm. White arrows highlight morphological changes in biofilms after treatment in comparison to the control.

**Figure 3 ijms-25-07769-f003:**
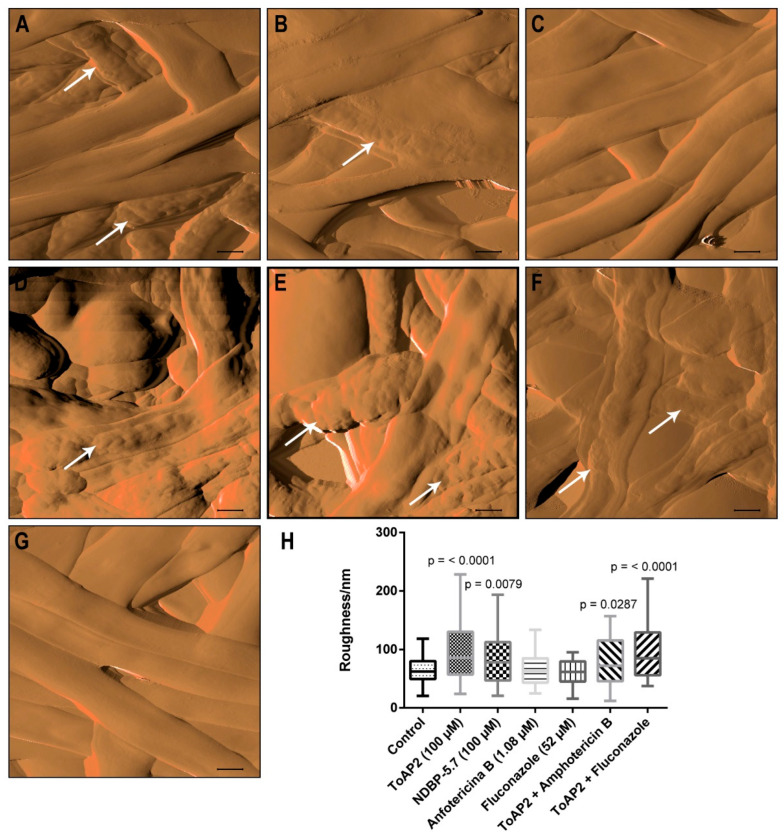
AFM amplitude images of *C. albicans* cells on mature biofilm after different treatments. (**A**) Treatment with 100 µM ToAP2, (**B**) 100 µM NDBP-5.7, (**C**) 52 µM Fluconazole, (**D**) 1.08 µM Amphotericin B, (**E**) combination of ToAP2 and Amphotericin, (**F**) combination of ToAP2 and fluconazole, (**G**) untreated control, (**H**) quantification of biofilm surface roughness after different treatments. Data were analyzed with one-way ANOVA and T-test (*p* < 0.05) Scale bar: 1 µm. Bars represent standard deviation. White arrows highlight morphological changes in biofilms after treatment in comparison to the control.

**Figure 4 ijms-25-07769-f004:**
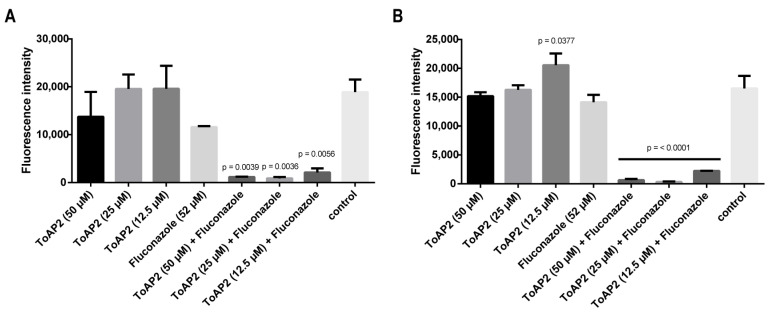
Viability of *C. albicans* cells on mature biofilms treated with different concentrations of ToAP2 or ToAP2 associated with Fluconazole. (**A**) Viability of *C. albicans* on mature biofilms formed in infusion tubes. (**B**) Viability of *C. albicans* mature biofilms formed in PU catheters. The data were analyzed by ANOVA (*p* < 0.05) and Tukey’s post-test. *p*-values of statistically significant comparisons are shown in the figure. Mean ± SEM.

**Figure 5 ijms-25-07769-f005:**
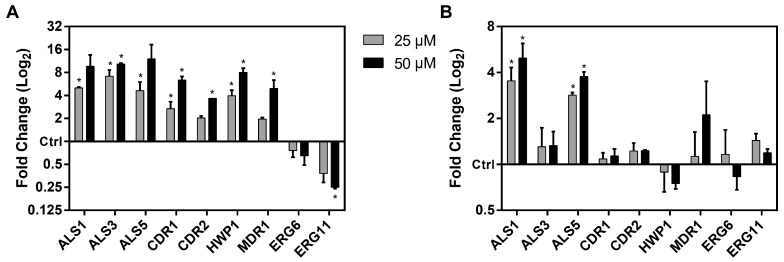
Effects of ToAP2 on the expression of genes related to *C. albicans* virulence factors. (**A**) Planktonic cells of *C. albicans* were treated with concentrations of 25 µM and 50 µM of ToAP2. (**B**) *C. albicans* biofilms were treated with concentrations of 25 µM and 50 µM of ToAP2. Data are presented as mean ± upper and lower limits. Statistical significance was calculated by Student’s *t*-test. * *p*-value < 0.05.

**Table 1 ijms-25-07769-t001:** Sequence of oligonucleotides.

Name	Gene	Primers	Reference
Actin	*ACT1*	F-CTGCTTTGGCTCCATCTTCTATG	
		R-AGCCAAGATAGAACCACCAATCC	
Hyphal cell wall protein 1	*HWP1*	F-CCGGAATCTAGTGCTGTCGT	[32,33]
		R-TAATTGGCAGATGGTTGCAT	
Agglutinin-like sequence protein 1	*ALS1*	F-CAGAGTTATGCCAAGTCTCAATAA	[59]
		R-CCCATTGTACCAGATGTGTAACCA	
Agglutinin-like sequence protein 3	*ALS3*	F-GTACTAGTGCAAGTCCGGGAGATA	[60]
		R-ACCATGAGCAGTCAAATCAACAGA	
Agglutinin-like sequence protein 5	*ALS5*	F-TGCCAATCCAGGGGATACATTC	[34]
		R-CACCATCGGCAGTCAAATCAAC	
Multidrug resistance	*MDR1*	F-TGGAGTTTGGGTGCTGTTTGTG	[37,38]
		R-AGTCCATCTCCAACTGGCTTTG	
*Candida* drug resistance 1	*CDR1*	F-TGCTGCCATGTTCTTTGCTG	[38]
		R-TCGACAATTGGTCTGGCTTCG	
*Candida* drug resistance 2	*CDR2*	F-GGGGTTGAATTAGTTGCCAAACC	[39]
		R-TCGACCAGGCAGTTTGAGAATC	
Sterol 24-C-methyltransferase	*ERG6*	F-AAGCTACCGTTCATGCTCCAG	[61]
		R-AAACACCGAAAACACCACCTG	
Sterol 14-demethylase	*ERG11*	F-ATTGGAGACGTGATGCTGCTC	[41]
		R-TGGATCAATATCACCACGTTCTC	

## Data Availability

Data is contained within the article and Appendix A.

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
