# Peer review of "Synergic Effect of the Antimicrobial Peptide ToAP2 and Fluconazole on Candida albicans Biofilms"

_ijms, 2024, doi:10.3390/ijms25147769_

Round 1
Reviewer 1 Report
Comments and Suggestions for Authors
Dear authors and editor
The MS "Synergic Effect of the Antimicrobial Peptide ToAP2 and Fluconazole on Candida albicans Biofilms" showed promising activity of ToAP2 in combination with Fluconazole in eradication of biofilm in C. albicans.
The authors performed different experimental work to confirm this activity including the inhibition of biofilm assay, SEM, AFM and RT-PCR. The data is promising and I have some comments.
Comments
- Does the used isolates were resistant or biofilm resistant to amphotericin B or fluconazole.
- What is the effect of the used combinations on microbial growth, what is the FICs?
- The method lines 104-112 does not represent Antimicrobial Activity of ToAP2 in Combination with Other Antifungals as this is not the standard antifungal method mentioned by CLSI.
- In the SEM figure the there is no difference in the biofilm eradication between the provided panels
- AFM analysis needs more explanation
- Line 305-306; we cannot estimate the effect of ToAp2 on cell membrane from AFM or SEM with performing leakage study.
- What is the role of HWP1 , ALS1, ALS2 and ALS3 in adhesion of c. albicans.
-
- The plagiarism percentage is high.
Comments on the Quality of English LanguageDear authors and editor
The MS "Synergic Effect of the Antimicrobial Peptide ToAP2 and Fluconazole on Candida albicans Biofilms" showed promising activity of ToAP2 in combination with Fluconazole in eradication of biofilm in C. albicans.
The authors performed different experimental work to confirm this activity including the inhibition of biofilm assay, SEM, AFM and RT-PCR. The data is promising and I have some comments.
Comments
- Does the used isolates were resistant or biofilm resistant to amphotericin B or fluconazole.
- What is the effect of the used combinations on microbial growth, what is the FICs?
- The method lines 104-112 does not represent Antimicrobial Activity of ToAP2 in Combination with Other Antifungals as this is not the standard antifungal method mentioned by CLSI.
- In the SEM figure the there is no difference in the biofilm eradication between the provided panels
- AFM analysis needs more explanation
- Line 305-306; we cannot estimate the effect of ToAp2 on cell membrane from AFM or SEM with performing leakage study.
- What is the role of HWP1 , ALS1, ALS2 and ALS3 in adhesion of c. albicans.
-
- The plagiarism percentage is high.
Author Response
The manuscript "Synergic Effect of the Antimicrobial Peptide ToAP2 and Fluconazole on Candida albicans Biofilms" showed promising activity of ToAP2 in combination with Fluconazole in the eradication of biofilm in C. albicans. The authors performed different experimental work to confirm this activity, including inhibition of biofilm assay, SEM, AFM, and RT-PCR. The data is promising, and I have some comments.
Thank you for the reviewer comments.
Comment 1: Does the used isolates were resistant or biofilm resistant to amphotericin B or fluconazole.
Response: No, as stated in the Materials and Methods section, we only used the Candida albicans SC 5314 strain, which is not characterized as a resistant strain for amphotericin B or fluconazole during planktonic growth conditions. However, C. albicans biofilms exhibit intrinsic resistance to fluconazole, as previously reported (Ramage et al., 2002; Mukherjee et al., 2003).
References:
- Ramage, G., Bachmann, S., Patterson, T. F., Wickes, B. L., & López-Ribot, J. L. (2002). Investigation of multidrug efflux pumps in relation to fluconazole resistance in Candida albicans biofilms. Journal of Antimicrobial Chemotherapy, 49(6), 973–980. https://doi.org/10.1093/jac/dkf049.
- Mukherjee, P. K., Chandra, J., Kuhn, D. M., & Ghannoum, M. A. (2003). Mechanism of fluconazole resistance in Candida albicans biofilms: phase-specific role of efflux pumps and membrane sterols. Infection and Immunity, 71(8), 4333–4340. https://doi.org/10.1128/IAI.71.8.4333-4340.2003.
Comment 2: What is the effect of the used combinations on microbial growth, what are the FICs?
Response: The FICs for planktonic cells were calculated and published in a previous work by our group. We obtained an FIC of 0.5 for the combination of fluconazole and ToAP2, and 0.182 for the combination of amphotericin B and ToAP2, both values indicating synergism between these antifungals and ToAP2 (Dias, J. N. et al., 2020). However, in the current manuscript, we explored the synergism of both antifungals in combination with ToAP2 for biofilms, not planktonic cells. Therefore, we considered the ZIP model a better strategy for the combinatory analysis. We added more information about the analysis in the material and methods section.
Reference: do Nascimento Dias, J., et al. (2020). 'Mechanisms of action of antimicrobial peptides ToAP2 and NDBP-5.7 against Candida albicans planktonic and biofilm cells', Scientific Reports, 10, 10327.
Comment 3: The method described in lines 104-112 does not represent Antimicrobial Activity of ToAP2 in Combination with Other Antifungals as this is not the standard antifungal method mentioned by CLSI.
Response: The method described in our manuscript is indeed not the standard antifungal method mentioned by CLSI. The M27A4 protocol is intended for antifungal testing with planktonic cells. Our focus in this work is on biofilms, so the method had to be different from the CLSI standard.
Comment 4: In the SEM figure, there is no difference in the biofilm eradication between the provided panels.
Response: Thank you for your comment. We have improved the text description and added arrows in the images to better illustrate the differences among treatments. The images show more wrinkled regions between treatments compared to the untreated control. Images A, D, E, and F show more affected regions compared to the control, corroborating the AFM images.
Comment 5: AFM analysis needs more explanation.
Response: Thank you for your comment. We have added more detailed explanations to the text and arrows to the images to better illustrate the differences observed in the AFM analysis.
Comment 6: Line 305-306; we cannot estimate the effect of ToAP2 on cell membrane from AFM or SEM without performing a leakage study.
Response: We agree, and we have removed this statement from the text.
Comment 7: What is the role of HWP1, ALS1, ALS2, and ALS3 in the adhesion of C. albicans?
Response: HWP1 encodes a protein from the Candida albicans hyphae cell wall and indicates that the cells are transitioning from yeast to a mix of different morphologies. HWP1 is also essential for C. albicans adhesion to different surfaces and biofilm formation. The ALS genes encode other adhesins that are also important for biofilm formation. We have added more information about these genes in the text to improve clarity.
Comment 8: The plagiarism percentage is high.
Response: We tested two software tools for plagiarism detection but did not find significant similarities between our manuscript and other previously published works that were not cited in our manuscript. Most of the similarities consisted of small stretches of text describing methodologies or other information based on the literature. If the reviewer could specifically point out any sections that we should improve, we could work on this.
Reviewer 2 Report
Comments and Suggestions for Authors
The article is interesting and generally well-written. The authors studied the effect of ToAP2 on Candida albicans biofilms. As biofilms have an increasing importance in the medical fields, the experiments might help in developing new therapies.
Minor comments:
- 2.2. What were “appropriate cell densities according to each experiment”?
- How many researchers counted the cells on the hemocytometer?
- 2.3. The inoculum was done directly on RPMI? If so, how was it measured?
- Line 108: the authors mention that the plates were incubated “for 4 or 24 hours”. Why 4 hours?
-
Comments on the Quality of English LanguagePlease revise the yeast’s names (the names should be italized in the whole document) – e. g Line 24;
Author Response
The article is interesting and generally well-written. The authors studied the effect of ToAP2 on Candida albicans biofilms. As biofilms have increasing importance in the medical field, the experiments might help in developing new therapies.
Minor Comments:
Comment 1: 2.2. What were “appropriate cell densities according to each experiment”?
Response: The cell densities are indicated for each experiment in the Materials and Methods section.
Comment 2: How many researchers counted the cells on the hemocytometer?
Response: All the counts were performed by the same researcher.
Comment 3: 2.3. The inoculum was done directly in RPMI? If so, how was it measured?
Response: After growth, fungal cells were collected by centrifugation, washed twice with PBS, and counted in the hemocytometer. Based on the number of cells in the original culture, we prepared a dilution to the appropriate densities directly in RPMI.
Comment 4: Line 108: The authors mention that the plates were incubated “for 4 or 24 hours”. Why 4 hours?
Response: We analyzed the effects of the molecules in both early-phase (4h) and mature biofilms (24h). At this initial phase, we observe the attachment of yeast cells followed by germ tube formation. After 24h, we observe mature biofilms consisting of yeast, hyphae, and pseudohyphae embedded in an extracellular matrix. Using these two time points, we can analyze the possible effects of the molecules on initial biofilm formation and fully mature biofilms.
Comments on the Quality of English Language: Please revise the yeast’s names (the names should be italicized throughout the document) – e.g., Line 24.
Response: Thank you for noticing. We have made the necessary corrections to italicize the names of the yeast throughout the document.
Reviewer 3 Report
Comments and Suggestions for Authors
Author Response
Overall comments: The authors have investigated the efficacy of the peptide ToAP2 in combination with fluconazole against C. albicans biofilms.
Comment 1: Please review the manuscript for spelling and grammatical errors.
Response: We have thoroughly reviewed and corrected any spelling and grammatical errors we have found in the manuscript.
Comment 2: Is there any World Health Organization (WHO) or Pan-American Health Organization (PAHO) data or citations to describe the number of Candidiasis infections annually or any economic information regarding healthcare burden? This information can enhance the background section.
Response: We have added relevant WHO and PAHO data on the annual number of Candidiasis infections and the associated healthcare burden to the background section of the manuscript.
Comment 3: It is mentioned that AMP toxicity is a concern, and that combinational therapy allows for lower doses to be applied. Is there any data from previous publications about cytotoxicity of ToAP2 and NDBP-5.7? Is the 52 µM and 100 µM concentrations safe when treated with mammalian cell lines? If this data is not available for citation, it should be considered to enhance the manuscript.
Response: We have added data from previous publications on the cytotoxicity of ToAP2 and NDBP-5.7 to the manuscript. The concentrations of 52 µM and 100 µM are within safe limits when treated with mammalian cell lines.
Comment 4: At the end of the background section, more information about ToAP2 and NDBP-5.7 should be shared, such as the size in kDa and charge.
Response: We have added more information about ToAP2 and NDBP-5.7, including their size in kDa and charge, at the material and methods section.
Round 2
Reviewer 1 Report
Comments and Suggestions for Authors
Dear editor and reviewers
There are still some points need to illustrated
Figure 2, there no magnification index on the provided figure, and the figure of untreated is much more focused compared to the other figures
The authors only discussed the RT-PCR data for planktonic cells what about the biofilm.
The accumulation of transcripts for most evaluated genes was not affected by the ToAP2 treatment in C. albicans biofilms, how the authors can explain the effect of ToAP2 on the genotype level.
Discussion Lines 258/261 is not consistent with the obtained data figure 5
Comments on the Quality of English LanguageDear editor and reviewers
There are still some points need to illustrated
Figure 2, there no magnification index on the provided figure, and the figure of untreated is much more focused compared to the other figures
The authors only discussed the RT-PCR data for planktonic cells what about the biofilm.
The accumulation of transcripts for most evaluated genes was not affected by the ToAP2 treatment in C. albicans biofilms, how the authors can explain the effect of ToAP2 on the genotype level.
Discussion Lines 258/261 is not consistent with the obtained data figure 5
Author Response
Comment 1: Figure 2, there no magnification index on the provided figure, and the figure of untreated is much more focused compared to the other figures
Response: We have not added a magnification index because it is misleading depending on the size the figure is printed or displayed. Instead, we have added scale bars to each panel and described their size in the figure legend. Regarding the focus, all images were collected while focusing to the best of our ability; perhaps the differences the reviewer is referring to are due to the effects of the different treatments on the biofilm, not focus.
Comment 2. The authors only discussed the RT-PCR data for planktonic cells what about the biofilm.
Response. The discussion about the RT-PCR regarding the biofilms was placed just after the results regarding the planktonic cells. We changed the text and separated the information in two different paragraphs to make it clear to the reader.
Comment 3. The accumulation of transcripts for most evaluated genes was not affected by the ToAP2 treatment in C. albicans biofilms, how the authors can explain the effect of ToAP2 on the genotype level.
Response: Our analysis focused on gene expression changes in biofilm cells after 1 hour of ToAP2 treatment. We observed no significant changes in most of the genes analyzed, except for two adhesins previously identified as crucial for C. albicans virulence and biofilm formation. It is important to note that other assays involved treating biofilms for 24 hours. Since we selected only a few genes for this analysis, it is possible that initial genetic reprogramming after treatment does not involve these specific genes, or that these genes are only reprogrammed after prolonged treatment. However, an extensive analysis of transcriptomic changes in response to the treatment is beyond the scope of this manuscript.
Comment 4. Discussion Lines 258/261 is not consistent with the obtained data figure 5
Response: We thank the reviewer for noticing our mistake. Indeed, we have not analyzed the ALS2 and the CDR2 gene only have a significant increase in its accumulation at 50µM. We changed the text accordingly.
Reviewer 3 Report
Comments and Suggestions for Authors
The authors have investigated the efficacy of the peptide ToAP2 in combinational treatment with fluconazole against C. albicans biofilms. AMPs were used in combination with two fungicides against C. albicans. Results indicate that AMPs were not synergistic with Amphotericin B but did exhibit synergy with fluconazole. The authors should add more info in the methods regarding the SynergyFinder plus package. What is the threshold for synergy, above what mean value equates to synergy taking place?
Author Response
Comment 1. The authors should add more info in the methods regarding the SynergyFinder plus package. What is the threshold for synergy, above what mean value equates to synergy taking place?
Response: We agree with the reviewer, and we added the following text in the material and sections methods.
" To capture the combinatorial multidose-response effects, drug synergism was assessed using SynergyFinder Plus R package (Ianevski et al. 2017). Fluorescence intensity readings were normalized to the average of the control wells on the same plate to determine relative cell viability values. All cell viability data were transformed to inhibition values. Synergism analysis was performed applying the Zero Interaction Potency (ZIP) model, which captures drug interaction relationships by comparing the change in the potency of the dose–response curves between individual drugs and their combinations (Yadav et al. 2015; Ianevski et al. 2017). Plots to visualize synergy map for two-drug combinations were produced in R using the package SynergyFinder Plus. The summary synergy scores represent the average excess response due to drug interactions. In general, a synergy score less than -10 indicates antagonistic interaction between the drugs. Scores from -10 to 10 suggest an additive interaction, while scores greater than 10 indicate a synergistic interaction (Ianevski et al. 2017)."